# Blood–Brain Barrier Disruption in Schizophrenia: Insights, Mechanisms, and Future Directions

**DOI:** 10.3390/ijms26030873

**Published:** 2025-01-21

**Authors:** Fangsong Zhang, Jianye Zhang, Xuexue Wang, Mengyang Han, Yi Fei, Jinhong Wang

**Affiliations:** Department of Medical Imaging, Shanghai Mental Health Center, Shanghai Jiao Tong University School of Medicine, 600 Wan Ping Nan Road, Shanghai 200030, China; zhangfs1505@smhc.org.cn (F.Z.); mmhh123456@smhc.org.cn (J.Z.); xuehua0117@smhc.org.cn (X.W.); hmy890930@smhc.org.cn (M.H.); feiyi19930919@smhc.org.cn (Y.F.)

**Keywords:** schizophrenia, blood–brain barrier, neuroinflammation, tight junction, endothelial dysfunction, neuroimaging, MRI, ASL, pathophysiology, biomarkers

## Abstract

The blood–brain barrier (BBB) plays a crucial role in maintaining the homeostasis of the central nervous system by regulating solute transport and preventing neurotoxic substances from infiltrating brain tissue. In schizophrenia, emerging evidence identifies BBB dysfunction as a key pathophysiological factor associated with neuroinflammation, tight junction abnormalities, and endothelial dysfunction. Recent advancements in neuroimaging techniques, such as arterial spin labeling (ASL), have provided valuable tools for investigating BBB permeability and its role in disease progression. This review synthesizes findings from postmortem studies, serum and cerebrospinal fluid biomarker analyses, and advanced neuroimaging research to elucidate BBB alterations in schizophrenia. It highlights the mechanistic roles of tight junction protein dysregulation, neurovascular unit dysfunction, and immune responses in disrupting BBB integrity. Furthermore, the review examines the bidirectional effects of antipsychotic medications on BBB, addressing both therapeutic opportunities and potential challenges. By emphasizing the pivotal role of BBB dysfunction in schizophrenia pathogenesis, this review underscores its translational potential. Through the integration of multidisciplinary evidence, it lays the foundation for innovative diagnostic approaches and therapeutic strategies, enhancing our understanding of schizophrenia’s complex pathophysiology.

## 1. Introduction

Schizophrenia (SZ) is a psychiatric disorder characterized by evolving clinical symptoms and treatment challenges, affecting approximately 24 million people worldwide. It commonly manifests in adolescence or early adulthood, often with a gradual onset. Major clinical symptoms include delusions, hallucinations, affective blunting, social withdrawal, and cognitive deficits [1]. Despite decades of research on the pathophysiological basis of schizophrenia, its fundamental pathogenic mechanisms remain unclear.

Genome-wide association studies (GWAS) indicate that schizophrenia has a polygenic basis, including some rare variants with significant effects [2] and hundreds of common genetic variants with small effects [3]. Genetic risk is more predictive of schizophrenia when adverse factors are present early in development. Therefore, schizophrenia is a complex disorder, with its onset and progression arising from the combined influence of genetic susceptibility and environmental factors [4,5,6].

The brain is known to be a high-energy- and high-oxygen-demanding organ. To meet these requirements, it requires substantial blood supply, resulting in the synchronous development of vascular and neural networks [7]. The total surface area of microvessels in the brain is approximately 150–200 cm^2^/g of tissue, equivalent to about 15–20 m^2^ for the entire adult brain, forming a dense vascular network supplying blood flow to all brain regions [8]. Given the lack of regenerative ability in neurons, maintaining the stability of the central nervous system (CNS) environment is crucial for both optimal synaptic transmission and safeguarding neuronal health and integrity [9]. Consequently, the brain vasculature differs from that of other body regions because of the presence of the blood–brain barrier (BBB), which serves as the primary defense of the brain against potentially noxious substances [10]. Recently, the BBB has emerged as a crucial dynamic interface for the brain and a key focus in research on the pathogenesis of various CNS disorders. Recent research has established that BBB dysfunction lies at the heart of the pathophysiology of multiple CNS diseases, including Alzheimer’s disease, chronic traumatic encephalopathy (CTE) [11], traumatic brain injury (TBI) [12], stroke, multiple sclerosis, epilepsy, and dementia [8]. Its role in psychiatric disorders, particularly schizophrenia, has gradually caught the attention of experts and scholars. Recent studies have revealed that disruptions in BBB function may have profound impacts on brain health and could play a significant role in the development and progression of psychiatric illnesses [13].

Studies have centered on structural and morphological abnormalities, changes in molecular marker expression, gene expression patterns, and immunological aberrations. The gold standard for detecting blood–brain barrier damage using peripheral and cerebrospinal fluid markers is the cerebrospinal fluid/serum albumin ratio [14]. Additionally, brain-derived neuronal growth factors (like S100 calcium-binding protein (S100B)) [15], and molecules associated with neurodegeneration, such as spectrin breakdown products 120 (SBDP120) and 145 (SBDP145), are potential biomarkers for BBB leakage [16]. Early neuroimaging studies were constrained by technology, but recent applications of techniques like DCE-MRI and ASL have enabled more in-depth observations of blood–brain barrier changes. The main observed indicators are the transfer constant (Ktrans) and water exchange rate (Kw), although the mechanisms of Kw and Ktrans may differ. Ktrans presumably represents abnormal passive permeability of water or macromolecules through the endothelium. A decrease in Kw may predominantly reflect impaired water transport across the neurovascular unit, which can signify early microscopic changes in the blood–brain barrier and is related to the function of aquaporin-4 (AQP-4) [17].These studies contribute to our understanding of the alterations in blood–brain barrier permeability in schizophrenia patients and lay the foundation for further investigations into its role in the pathogenesis.

However, although previous studies have yielded certain results, research on the mechanisms underlying BBB disruption remains in the exploratory phase, with most studies concentrating on neuroinflammation, neurovascular unit dysfunction, and tight junction leakage. Disruption of blood–brain barrier (BBB) integrity, caused by the downregulation of tight junction or adherens junction proteins, can lead to compromised barrier function by increasing overall permeability (leakage) in brain endothelial cells. Certain proteolytic enzymes, such as Matrix Metalloproteinase 9 (MMP9), are known to degrade the basement membrane of blood vessels and tight junction proteins. Consequently, the upregulation of MMP9 can undermine BBB integrity and enhance its permeability [18,19].

When systemic inflammation occurs, brain endothelial cells (BECs) become activated, leading to an increase in BBB’s selective permeability to immune cells from the bloodstream. This is mediated by the upregulation of adhesion molecules like Vascular and Intercellular Cell Adhesion Molecules (VCAM1 and ICAM1), as well as P-selectin and E-selectin [20,21,22]. Considering these findings, a BBB that shows heightened sensitivity to environmental stressors, potentially due to inherent alterations in BEC functionality in schizophrenia (SZ), may allow the entry of harmful substances and peripheral immune cells into the central nervous system (CNS), thereby triggering neuroinflammation. On the other hand, elevated systemic inflammation might directly contribute to the increased BBB permeability observed in SZ patients.

Although research on BBB dysfunction in schizophrenia (SZ) has made some headway, advancements in new technologies, such as dynamic contrast-enhanced magnetic resonance imaging (DCE-MRI) and arterial spin labeling (ASL), have provided novel tools for studying the integrity and function of the BBB in SZ. However, significant knowledge gaps remain regarding the specific mechanisms driving BBB alterations, their causal relationship with SZ symptoms, and the translational potential of these findings for diagnosis and treatment.

In this review, we integrate multidisciplinary evidence to summarize the existing mechanisms of blood–brain barrier (BBB) dysfunction in schizophrenia (SZ). Our aim is to deepen the understanding of how BBB dysfunction contributes to the pathophysiology of SZ and to explore the causal relationships between BBB alterations and SZ symptoms. Special emphasis on a detailed introduction to state-of-the-art detection techniques, such as arterial spin labeling (ASL), is provided. Meanwhile, the limitations of current studies are critically evaluated, and future research directions are proposed in this article.

## 2. Structure and Function of the Blood–Brain Barrier

The BBB, which surrounds the central nervous system, is one of the few highly selective and tightly regulated barriers, highlighting the brain’s critical role in cognitive function, homeostasis, and the coordination of peripheral organ functions. The BBB, a part of the neurovascular unit (NVU), represents a complex structure comprising diverse cell types, including neurons, astrocytes, microglia, pericytes, and brain microvascular endothelial cells (see Figure 1A). The cell types within the NVU work together to ensure a steady blood supply to the brain, a process known as autoregulation. Whenever neuronal activation leads increased metabolic demands, cerebral blood flow promptly rises [23]. In fulfilling such functions, the BBB is of vital importance in maintaining the brain’s microenvironmental homeostasis. Through tight junctions among endothelial cells, receptor- and transporter-mediated protein and peptide transportation, and transcytosis across endothelial cells, the BBB regulates the exchange of ions and nutrients between the blood and the brain. It also safeguards vulnerable neural tissue from potentially hazardous blood-borne substances such as pathogens, immune cells, and bradykinin [24]. Additionally, brain endothelial cells secrete approximately 200 milliliters of new interstitial fluid daily, contributing in maintaining the optimal ionic environment necessary for normal neural function [25].

The junctional complexes, which include adherens junctions and tight junctions, are located within the gaps between brain microvascular endothelial cells. Together with the basement membrane, pericytes, and astrocyte endfeet that form the glial limiting membrane, these complexes construct the blood–brain barrier (BBB), restricting its permeability [7]. Adherens junctions consist of cadherin and catenin proteins spanning the endothelial cell gaps. Catenins further bind to cytoskeletal proteins, linking cells and regulating intercellular connections. Although the exact role of adherens junctions is not yet fully elucidated, they are thought to contribute to maintaining cell polarity, providing stability, promoting endothelial cell survival, and responding to stimuli via interactions with cadherin or catenin proteins and the actin cytoskeleton [26]. Unlike adherens junctions distributed throughout all vascular beds, tight junctions are highly enriched in the cerebral microvascular endothelium. Tight junctions span the intercellular spaces and interact with tight junction proteins on adjacent endothelial cells at the “contact points”, thereby sealing the intercellular space [27]. The binding of tight junction proteins impedes the flow of solutes and ions between the blood and the brain, thus forming a dynamic and highly regulatable barrier system. Tight junctions are mainly composed of claudins and occludins, which are connected to the actin cytoskeleton through zonula occluden (ZO) proteins [28]. The presence of tight junctions restricts the intercellular permeability and maintains cell polarity by achieving the asymmetric distribution of membrane components [29].

In addition to endothelial cells, pericytes, astrocytes, and the basement membrane (BM) are also crucial elements in maintaining the function and integrity of the blood–brain barrier (BBB). The presence of astrocytic endfeet encircling blood vessels is a characteristic feature of the brain vasculature, emerging during the development of the BBB. Astrocytes can partly activate the barrier function of endothelial cells and are significant in BBB maintenance. Studies have shown that the existence of astrocytes is one of the prerequisites for the formation and maturation of tight junctions (TJs) between brain endothelial cells [30,31]. Moreover, astrocytes secrete several regulatory factors, including TGF-β, GDNF, bFGF, and IL-6 [32]. The basement membrane is a specialized extracellular matrix. Its biological activity is essential for the growth, development, differentiation, and functional maintenance of endothelial cells. Composed mainly of fibronectin (FN), type IV collagen, laminin (LN) linked to collagen, and proteoglycans, the basement membrane helps in positioning cells and establishes connections among endothelial cells, astrocytes, and pericytes. It also functions as an additional barrier to solute movement prior to entering neural tissue [33]. Damage to the basement membrane results in the dysregulation of tight junction protein expression on endothelial cells, leading to changes in blood–brain barrier permeability [34].

Pericytes are mural cells embedded in the basement membrane, surrounding the blood vessels of the central nervous system (CNS) during embryonic development. They are essential for the establishment of the blood–brain barrier (BBB) and are attracted to endothelial cells by the paracrine signaling of platelet-derived growth factor β (PDGFβ) and its receptor (PDGFRβ). Indeed, PDGFβ and PDGFRβ knockout mice die at the embryonic stage, exhibiting tight junction dysfunction and enhanced vascular permeability as a result of the absence of pericyte coverage. Pericytes modulate BBB permeability by governing the expression of tight junction proteins and adherens junction proteins. In addition, pericytes can engage in immune responses with macrophage-like activity. They express scavenger receptors, and cultured pericytes possess the capacity to phagocytose large molecules like polystyrene beads. However, the role of pericytes in psychiatric disorders has yet to be thoroughly investigated thus far [13,35,36]

In summary, the blood–brain barrier (BBB) is a dynamic boundary between the brain and the surrounding tissues, and the self-regulation of the neurovascular unit (NVU) plays a critical role in maintaining normal physiological functions of the brain. Endothelial cells strictly regulate the movement of substances into and out of the brain through tight junctions, maintaining the selective permeability of the blood–brain barrier. Astrocytes, through their foot processes, are tightly connected to endothelial cells, and they not only deliver nutrients and metabolic products but also dynamically regulate the expression and function of tight junction proteins in endothelial cells through the release of cytokines such as transforming growth factor-β (TGF-β), ensuring the integrity of the blood–brain barrier. Smooth muscle cells encircle blood vessels and, based on neuronal activity and local metabolic demands, secrete signaling molecules such as nitric oxide (NO) or endothelin to precisely regulate vascular tone and blood flow, ensuring adequate blood supply to the brain. Microglia constantly monitor the central nervous system environment and are rapidly activated under inflammatory stimuli [37]. They collaborate with endothelial cells and astrocytes by secreting cytokines such as tumor necrosis factor-α (TNF-α) and interleukin-1 (IL-1) to regulate neuroinflammatory responses and prevent excessive inflammation from damaging the neurovascular unit. Neurons regulate vasodilation and vasoconstriction bidirectionally through the release of neurotransmitters and neurotrophic factors, such as brain-derived neurotrophic factor (BDNF), while also influencing the metabolism and function of astrocytes and endothelial cells, maintaining the homeostatic balance of the neurovascular unit [21]. Under physiological conditions, these cell types work closely together and synergistically, forming a highly precise and dynamically balanced self-regulating system that effectively resists external disturbances and maintains the stability of the brain’s microenvironment [38]. In pathological conditions, such as neurodegenerative diseases, cerebrovascular disorders, or neuroinflammation, the self-regulatory mechanisms of the neurovascular unit may be impaired, leading to an imbalance in cellular interactions. This results in blood–brain barrier dysfunction, abnormal blood flow perfusion, and exacerbated neuroinflammation, further aggravating disease progression.

## 3. Methodology-Based Studies on Blood–Brain Barrier Permeability in Schizophrenia

### 3.1. Post-Mortem Studies

Postmortem studies have been increasingly carried out, mainly centering on structural and morphological abnormalities, alterations in molecular marker expression, gene expression patterns, and immunological abnormalities. However, these studies are frequently constrained by small sample sizes, inadequate control group matching, the potential confounding effects of medications, variations in sample preservation conditions, and biases arising during brain tissue processing, all of which could adversely affect the results.

#### 3.1.1. Structural and Morphological Anomalies

In the prefrontal cortex and visual cortex of patients with schizophrenia, morphological differences in capillaries and NVU (neurovascular unit) cell types have been observed, along with endothelial cell vacuolation, astrocytic end-foot thickening, and basal membrane thickening [39]. Subsequent studies have found that a decrease in capillary density correlates with the negative symptoms of schizophrenia [40]. Structural abnormalities of varying extents have also been noted in the brains of schizophrenia patients under antipsychotic treatment, such as reduced capillary diameter, extracellular matrix deposition, perivascular edema, and endothelial cell vacuolation [41]. However, this particular study only included three patients on antipsychotic treatment. Other studies have pointed to abnormalities in other NVU components, particularly astrocytes. A reduced number of oligodendrocytes surrounding capillaries in the prefrontal cortex of schizophrenia patients has been reported [42]. Moreover, a decreased quantity of glial fibrillary acidic protein (GFAP)-positive astrocytes around blood vessels has been observed in the prefrontal cortex, hippocampus, and anterior cingulate cortex of schizophrenia patients [43,44]. The extent to which these anomalies influence the blood–brain barrier function remains unclear.

#### 3.1.2. Altered Expression of Molecular Markers

A study used immunohistochemical techniques to assess P-glycoprotein expression in the brains of 13 patients with schizophrenia and 9 healthy controls. They found no significant differences in the density of P-glycoprotein-expressing capillaries in most regions; however, a decreased capillary density was observed in the habenula of schizophrenia patients. Neurons in the habenula also express P-glycoprotein, and schizophrenia patients showed a reduced density of P-glycoprotein-expressing neurons [45]. Another study reported a significant reduction in claudin-5 in the hippocampus of schizophrenia patients. Moreover, the mRNA and protein levels of claudin-5, claudin-12, and ZO-1 were found to be associated with the age of onset and the duration of schizophrenia [46].

#### 3.1.3. Gene Expression Abnormalities

A study performed single-nucleus RNA sequencing on midbrain tissues from 15 psychiatric patients and 14 healthy controls. The results indicated no alterations in the relative abundance of major blood–brain barrier (BBB) cell types associated with schizophrenia, nor were there any alterations in subpopulations linked to the disorder. However, 14 differentially expressed genes were identified in BBB cells of schizophrenia patients compared to controls, including previously implicated schizophrenia-related genes such as FOXP2 and PDE4D. These transcriptional changes were restricted to ependymal cells and pericytes, suggesting that such alterations are not widespread in schizophrenia [47].

Considering the diverse causes and symptom presentations in schizophrenia (SZ), researchers utilized a statistical approach to analyze gene expression variability. Their analysis revealed a general increase in gene expression variability across SZ brains, with VEGFA emerging as the most variably expressed gene compared to controls [48]. This discovery provides a unifying explanation for the inconsistent findings across studies regarding VEGF’s involvement in SZ. The variability in VEGF expression among individuals with SZ suggests it may play distinct roles at different stages of the disease.

BAI3 (Brain-specific Angiogenesis Inhibitor 3) plays a critical role in the process of angiogenesis [49]. Recent postmortem findings revealed a male-specific decrease in hippocampal BAI3 expression among SZ patients, based on an analysis of 104 SZ cases and 174 controls [50].

#### 3.1.4. Immunological Abnormalities

A postmortem study quantitatively evaluated the presence of lymphocytes in the brain parenchyma of schizophrenia patients. The study detected the significant infiltration of CD3+ T lymphocytes and CD20+ B lymphocytes in the hippocampal region in residual and paranoid schizophrenia, but no correlation with disease duration was observed [51]. Another study, which partially corroborated these findings, investigated IgG levels in the brain parenchyma of schizophrenia patients. It showed elevated IgG levels in the temporal cortex surrounding the hippocampus and in the hippocampus itself compared to controls, suggesting increased BBB permeability in these regions. Additionally, blood–brain barrier dysfunction in schizophrenia patients was more prominent in older and male individuals [52].

A study categorized schizophrenia (SZ) postmortem cases into high-inflammation and low-inflammation subtypes based on inflammatory cytokine transcript levels in brain tissue. The SZ-high-inflammation group displayed significantly elevated midbrain ICAM and macrophage marker CD163 expression compared to controls, while no differences were observed in the SZ-low-inflammation group [53]. Further analysis of dorsolateral prefrontal cortex (DLPFC) endothelial transcripts revealed the altered expression of key blood–brain barrier (BBB) markers in the SZ-high-inflammation group, including increased CDH5 and OCLN and decreased ABCG2. Additionally, SZ-high-inflammation samples exhibited increased ICAM1 expression and a higher number of CD163+ macrophages. These findings suggest that neuroinflammation, endothelial dysfunction, increased BBB permeability, and greater peripheral macrophage infiltration are associated with the high-inflammation SZ subtype. Importantly, ICAM and other BBB gene expressions in cultured brain endothelial cells were not affected by therapeutic doses of antipsychotics [54].

SERPINA3 has anti-angiogenic and anti-inflammatory effects on endothelial cells, and its expression is increased in the cerebral cortex of SZ patients in postmortem analyses [55]. Recent studies have verified the elevated expression of SERPINA3 in SZ patients, with even higher levels observed in the high-inflammation SZ group. Immunostaining revealed that blood vessel-associated astrocytes are the primary source of increased SERPINA3 expression in the high-inflammation SZ group. This upregulation may constitute a compensatory response to chronic inflammation in SZ patients [56]. SERPINA3 inhibits the catalytic activity of leukocyte elastase, an enzyme that cleaves ICAM1 and facilitates leukocyte adhesion to endothelial cells. Therefore, the increased SERPINA3 expression in blood vessel-associated astrocytes of the high-inflammation SZ group may act in concert with CD163+ CNS infiltration [54].

### 3.2. Peripheral and Cerebrospinal Fluid Markers

The gold standard for assessing BBB permeability in humans is the cerebrospinal fluid (CSF) to serum albumin ratio (QAlb). This test gauges BBB permeability by comparing the albumin concentration in CSF and blood. Albumin concentration in CSF is typically about 200 times lower than in blood; thus, an elevated QAlb indicates increased albumin leakage from blood into CSF due to barrier damage. This test has been utilized in several psychiatric studies to assess BBB dysfunction [57]. In a study involving 63 psychiatric patients and 4100 controls, some psychiatric patients (14 with major depressive disorder and bipolar disorder and 14 with schizophrenia) showed abnormal CSF mirroring-altered BBB permeability, p chiefly characterized by increased QAlb levels [58]. BBB dysfunction is also observed in diverse forms of dementia, like Alzheimer’s disease and frontotemporal dementia, with elevated QAlb levels reported in these patients [59]. A recent study contrasted 104 first-episode psychosis (FEP) patients with 104 healthy controls regarding CSF white blood cell (WBC) counts, total protein, IgG index, and CSF/serum albumin ratio. The results evinced that schizophrenia patients had increased CSF/serum albumin and CSF/serum IgG ratios compared to controls. More patients in the psychosis group had CSF WBC > 3 cells/µL than controls, along with elevated serum leukocyte counts and neutrophil-to-lymphocyte ratios. This suggests that when comparing a large sample of psychosis patients to healthy controls, increased BBB permeability, elevated CSF WBC levels, and enhanced peripheral inflammation are present when comparing a large sample of psychosis patients to healthy controls [60]. Reduced levels of vitamin B12 and folate can hike homocysteine levels, potentially contributing to BBB alterations. Using this as a basis, researchers analyzed routine CSF and blood parameters in a cohort of first-episode psychosis (FEP) patients to probe the relationship between vitamin B12/folate deficiency and BBB disruption. The study found that 17.1% (38/222) of patients had elevated CSF/serum albumin ratios (QAlb), 29.3% (62/212) had white matter lesions, and 17.6% (39/222) exhibited reduced vitamin B12 or folate levels. However, no statistically significant association was identified between vitamin deficiency and QAlb changes [61]. In a subsequent study, the relationship between CSF changes and C-reactive protein (CRP) levels was dissected. Among the cohort, 41.4% of patients had elevated CRP levels, but no significant association was found between CRP and QAlb changes [62]. Nevertheless, analyzing QAlb has limitations, as its elevation could be caused by other factors, including a reduced CSF production rate, increased subarachnoid space flow resistance, or impaired outflow due to arachnoid villi blockage. Additionally, this method is invasive and less practical for schizophrenia patients. Furthermore, QAlb is not a specific indicator and is susceptible to influences such as CSF production and circulation, making it an unreliable gauge of BBB permeability.

The calcium-binding peptide S100β is mainly produced by astrocytes and oligodendrocytes and is abundantly expressed in neurons and meninges in the brain. In healthy individuals, S100β is copiously detectable in serum. Consequently, elevated serum S100β concentrations have been used to correlate central nervous system (CNS) disorders, such as traumatic brain injury, with BBB dysfunction [63]. A growing number of studies have reported elevated S100β levels in the blood, cerebrospinal fluid (CSF), and brain tissues of schizophrenia patients [64,65]. Moreover, plasma S100β levels positively correlate with the negative symptoms [66,67] and cognitive deficits [68] associated with schizophrenia. A meta-analysis revealed that S100β concentration positively correlates with positive symptoms and is related to the course of psychosis, intimating the possibility that BBB permeability increases as the psychiatric disorder progresses [63]. However, it remains unclear whether the increase in S100β directly reflects enhanced BBB permeability or merely indicates increased production and/or secretion of glial cells or glial degeneration [69]. Additionally, S100β has been found to be secreted by adipose tissues outside the CNS, casting doubt on the interpretation and validity of S100β as a marker for BBB permeability [70].

Molecules associated with neurodegeneration, such as spectrin breakdown products 120 (SBDP120), 145 (SBDP145), and 150 (SBDP150), claudin, occludin, and ubiquitin C-terminal hydrolase-L1 (UCHL1), have been reported to be related to the blood–brain barrier (BBB). A study evaluated the levels of these peripheral biomarkers in healthy controls and two groups of schizophrenia patients (one receiving typical and atypical antipsychotic treatments and the other receiving only atypical antipsychotic treatment). The results showed that in schizophrenia patients treated with atypical antipsychotics, the levels of SBDP145 and cullin (a ubiquitin ligase) were significantly higher than those in the control group. In the combined treatment group, SBDP150 levels were lower than those in the control group (*p* = 0.022). Levels of claudin, occludin, and UCHL1 were similar between the two groups. Among patients receiving combined treatment, SBDP145 levels positively correlated with the total SAPS scores and SAPS delusion subscores. This study represents the first exploration examining the relationship between schizophrenia and BBB damage through the measurement of SBDP145, SBDP150, UCHL1, cullin, occludin, and claudin [71].

The endothelial glycocalyx (GLX) serves as a protective layer in the blood–brain barrier (BBB), and its degradation can lead to BBB dysfunction. A study utilized immunoblotting to measure 11 GLX biomarkers in the peripheral blood of first-episode schizophrenia patients and healthy controls. Compared to healthy controls, three GLX biomarkers were significantly elevated in patients. Moreover, the increase in GLX biomarkers was associated with symptom severity. This study demonstrated the potential of GLX molecules as immunoneuropsychiatric biomarkers for the early diagnosis of psychosis. Further research is required to explore the role of GLX in the early detection of psychotic disorders [72].

Peripheral and cerebrospinal fluid markers, such as the CSF/serum albumin ratio (QAlb) and neurodegenerative biomarkers, like S100β, are commonly used to assess BBB permeability in schizophrenia. While QAlb reflects BBB dysfunction, its specificity is limited by factors like CSF production and flow resistance. S100β, spectrin breakdown products, and GLX require further study to validate their specificity and sensitivity. Future research should focus on improving the specificity and sensitivity of these markers and consider combining them with non-invasive imaging methods to better assess BBB function.

### 3.3. Neuroimaging Studies

In previous studies, neuroimaging has predominantly focused on structural brain changes in schizophrenia patients, with consistent findings across multiple studies. Numerous studies have corroborated volume reductions in the medial temporal lobe (linked to memory) and the left posterior superior temporal gyrus (involved in auditory processing and language), as well as ventricular enlargement in schizophrenia patients [73]. However, using neuroimaging techniques to evaluate BBB changes in schizophrenia patients has been limited, and several early studies were hindered by small sample sizes, difficulties in result interpretation, and technical limitations. A magnetic resonance imaging (MRI) study using a 7T scanner reported changes in the volume of small arteries and arterioles in the brain’s vasculature, suggesting widespread microvascular abnormalities exist throughout the brain [74], potentially explicating the reduced gray matter volumes associated with schizophrenia. This study was predicated on previous findings of cerebral blood flow (CBF) and cerebral blood volume (CBV) abnormalities in schizophrenia [75]. Given the focus on smaller blood vessels, these findings may be relevant to BBB function. Advances in dynamic contrast-enhanced magnetic resonance imaging (DCE-MRI), a technique for quantitatively assessing BBB permeability, may facilitate the detection of subtle BBB abnormalities. A recent study leveraging this technology compared structural MRI and DCE-MRI data from 29 schizophrenia patients and 18 controls. The results revealed that schizophrenia patients exhibited higher Ktrans values bilaterally in the thalamus compared to healthy controls. The mean Ktrans value in the thalamus was significantly positively correlated with disease duration and symptom severity. Subregional analysis of the thalamus demonstrated pronounced BBB disruption in the pulvinar nucleus, particularly in the medial and lateral thalamic nuclei. Furthermore, Ktrans values across the whole brain, thalamus, and thalamic subregions were negatively correlated with their respective volumes [76].

Researchers employed DP-ASL technology to measure water exchange (Kw) between brain capillaries and neurons in schizophrenia patients and to evaluate changes in BBB permeability. Meanwhile, brachial artery flow-mediated dilation was used to assess peripheral vascular endothelial cell health, exploring the relationship between whole-brain Kw and endothelial function. The study divulged a significantly reduced mean whole-brain Kw in the patient group, with decreased neurovascular water exchange in the right parietal lobe, particularly in the supramarginal gyrus and postcentral gyrus. Additionally, reduced Kw in the right superior corona radiata and angular gyrus was associated with negative symptoms. Schizophrenia patients (SSD) also exhibited significantly impaired peripheral vascular endothelial cell function; in the healthy control (HC), 94% of brain regions showed a positive correlation between Kw and peripheral endothelial function, whereas in the SSD group, 52% of brain regions displayed a negative correlation [77]. Studies on DP-ASL and DCE-MRI have made the real-time, in vivo observation of BBB permeability possible, though the mechanisms of Kw and Ktrans may vary. Reduced Kw may mainly reflect impaired water transport across neurovascular units, indicating subtle early changes in the BBB, potentially related to the aquaporin-4 (AQP-4) function. In contrast, Ktrans may represent the abnormal passive permeability of water or macromolecules through the endothelium [78,79,80,81].

Certain central nervous system diseases, including epilepsy, are associated with the upregulation of the P-glycoprotein function, which may represent a compensatory mechanism triggered by blood–brain barrier permeability damage. A PET study attempted to quantify the function of blood–brain barrier transporters in vivo. The authors used radiolabeled verapamil ([^11^C]) as a probe to assess P-glycoprotein function and distribution in a small sample of schizophrenia patients. Compared to healthy controls, the decreased uptake of [^11^C]verapamil was observed in the temporal cortex, basal ganglia, and amygdala, indicating enhanced P-glycoprotein activity [82].

## 4. Mechanistic Studies on Blood–Brain Barrier Abnormalities in Schizophrenia

Currently, research into the mechanism of blood–brain barrier injury remains at the exploratory stage. In recent years, numerous studies have mainly focused on neuroinflammation, dysfunctions of the neurovascular unit, and tight junction leakage, among other aspects.

### 4.1. The BBB and Immune Response in Schizophrenia

As a part of the pathophysiology of schizophrenia, neuroinflammation is increasingly regarded as a significant factor in the pathogenesis of this disorder. Human and animal studies have shown that schizophrenia-related neuroinflammation can disrupt neurovascular function, leading to impaired blood–brain barrier function [21]. The main pathways involved include microglial activation and proliferation (MAP), upregulation of matrix metalloproteinases (MMPs), upregulation of pro-inflammatory cytokines, and activation and apoptosis of astrocytes.

Excessive microglial activation is associated with the pathophysiology of schizophrenia. Postmortem studies of the brains of schizophrenia patients consistently demonstrate elevated microglial activation. For example, there is increased HLA-DR immunoreactivity in multiple regions, particularly the dorsolateral prefrontal cortex, superior temporal cortex, and anterior cingulate cortex, compared with healthy controls [83,84]. In the hippocampal regions of some schizophrenia patients, the enrichment of CD3+ T lymphocytes, CD20+ B lymphocytes, and HLA-DR-expressing microglia has been observed [51]. Excessive microglial activation disrupts the tight junction proteins of BBB endothelial cells and increases BBB permeability through various mechanisms. These include the activation of inducible nitric oxide synthase, the promotion of reactive oxygen species (ROS) synthesis, inducting COX2 expression within the neurovascular unit, and the upregulation of pro-inflammatory cytokines and matrix metalloproteinases. Increased BBB permeability may, in turn, enhance interactions between innate brain immunity and peripheral adaptive immunity, perpetuating microglial activation and pro-inflammatory cytokine synthesis in the brain through a positive feedback loop [21].

Matrix metalloproteinase-9 (MMP9) is a 92 kDa protein that belongs to the family of zinc- and calcium-dependent endopeptidases. Studies have found elevated concentrations in schizophrenia patients, along with tissue inhibitors of metalloproteinases (TIMP) [85,86]. The primary mechanism by which MMP9 negatively impacts the central nervous system (CNS) likely involves its participation in immune-inflammatory responses. This includes activating and regulating various cytokines and chemokines, leading to the disruption of the blood–brain barrier (BBB). The resulting BBB breakdown allows leukocytes to infiltrate brain tissue, thereby exacerbating the adverse effects of neuroinflammation on the brain. However, the relationship between MMP9 upregulation, lipid peroxidation, and BBB disruption in schizophrenia patients remains uncertain.

Due to the enhanced “immune privilege” status conferred by the blood–brain barrier (BBB), alterations in BBB components such as cell adhesion molecules and vascular endothelial growth factors (VEGF) may increase the infiltration of peripheral immune cells into the brain. A few studies have examined BBB components in the peripheral blood of patients. Several have shown that untreated acute schizophrenia patients exhibit increased peripheral concentrations of endothelial cell adhesion molecules, such as P-selectin and L-selectin, in their serum and plasma compared to controls [87,88]. Changes in vascular endothelial growth factor (VEGF), intercellular adhesion molecule-1 (ICAM-1), and vascular cell adhesion molecule-1 (VCAM-1) levels have also been observed in schizophrenia patients [89]. A recent study reported elevated serum VEGF levels in schizophrenia patients, which were associated with structural abnormalities in the prefrontal cortex [90]. The accumulation of vascular endothelial factors may contribute to the increased transendothelial migration of inflammatory mediators, facilitating the aggregation of peripheral immune cells in the central nervous system and thereby inducing a neuroinflammatory state. This condition can lead to endothelial cell activation, subsequently resulting in abnormal BBB permeability.

A study has shown that the levels of pro-inflammatory cytokines in the blood and cerebrospinal fluid of schizophrenia patients are significantly elevated, with cytokine levels decreasing after the resolution of psychotic symptoms. Neuroinflammation activates microglial cells, which subsequently secrete cytokines such as interleukin (IL)-1α, IL-1β, IL-6, and tumor necrosis factor (TNF)-α, thereby impairing the endothelial function of the blood–brain barrier (BBB) [91]. The primary mechanisms by which pro-inflammatory cytokines contribute to BBB disruption may include inducing the expression of intercellular adhesion molecule-1 (ICAM-1) and vascular cell adhesion molecule-1 (VCAM-1) on the luminal surface of BBB endothelial cells, thereby promoting the transendothelial migration of lymphocytes and monocytes, causing oxidative damage to endothelial cells by impairing mitochondrial oxidative metabolism in vascular endothelial cells, and directly damaging the tight junctions of endothelial cells.

Astroglia regulate cerebral blood flow and volume as well as BBB permeability, among many other critical functions [92]. Therefore, in schizophrenia, there is a recorded reduction in the number of astroglia within functionally significant regions, including the subgenual cingulate, anterior, dorsolateral, and prefrontal cortices, along with the hippocampus and corpus callosum [84]. Aquaporin 4 (AQP4), a bidirectional water channel predominantly expressed at the perivascular endfeet of astrocytes, is crucial for BBB development, integrity, and intracerebral water homeostasis [93]. A significant reduction in AQP4 expression in the deep layers of the anterior cingulate cortex in schizophrenia patients has been confirmed [94]. Reduced AQP4 expression may impair astrocyte-endothelial interactions, resulting in neurovascular dysfunction and increased BBB permeability [95]. The pathological mechanism by which altered AQP4 expression impacts glial cell function and damages the BBB in schizophrenia patients requires further investigation.

A comprehensive review of the aforementioned studies suggests that neuroinflammation may contribute to the behavioral and cognitive symptoms of schizophrenia through multiple mechanisms. These mechanisms include disrupting the integrity of the blood–brain barrier (BBB), reducing cerebral blood flow, and disturbing the dynamic balance of the brain’s microenvironment. Furthermore, BBB disruption may exacerbate interactions between the brain’s innate immune system and the peripheral adaptive immune system, leading to chronic harmful neuroimmune signaling and neuroinflammatory responses. However, many of these findings remain speculative, underscoring the urgent need for further research to elucidate the precise relationship between these mechanisms and the pathogenesis and progression of schizophrenia.

### 4.2. The Abnormalities of the Neurovascular Unit in Schizophrenia

Neurovascular endothelial dysfunction is closely linked to increased blood–brain barrier permeability. Studies have shown that oxidative stress and neurovascular endothelial dysfunction in schizophrenia may arise from mechanisms such as impaired cerebral blood perfusion and disturbances in the homeostasis of the brain microenvironment. Furthermore, the disruption of blood–brain barrier permeability allows harmful neuroimmune signals and neuroinflammatory responses to persist, leading to the typical cognitive, emotional, and behavioral symptoms of schizophrenia [21].

In the prefrontal cortex and visual cortex of schizophrenia patients, studies have detected structural and morphological abnormalities in capillaries and neurovascular unit (NVU)-related cells. These include a decrease in capillary density, endothelial cell degeneration, and alterations in astrocytic endfeet and basement membranes [39,40,41]. These changes are associated with the negative symptoms of schizophrenia and antipsychotic treatment. However, the limited sample sizes in some studies demand further validation. Additionally, other NVU components, such as astrocytes, also display abnormalities [42,43,44]. These microvascular abnormalities are relevant to the pathogenesis of schizophrenia, but the degree of their impact requires further investigation.

### 4.3. The Tight Junction Leakage in Schizophrenia

CLDN5 (Claudin-5) is a key protein constituting the tight junctions of endothelial cells and serves as a primary transmembrane protein of the blood–brain barrier (BBB) [96]. It is a critical marker in studies of BBB structural changes, with its structure, distribution, and abnormal expression tightly associated with alterations in BBB permeability. Studies have shown that in the prefrontal cortex (non-visual cortex) of schizophrenia patients, CLDN5 mRNA expression is significantly increased while claudin-5 protein levels are decreased, with no apparent abnormalities in the visual cortex [97]. Transcriptomic analyses of the prefrontal cortex revealed that 12 out of 21 tight junction-related genes were decreased in schizophrenia patients compared to controls [98]. Another study reported contrary results, finding a significant reduction in hippocampal CLDN5 levels in schizophrenia patients, with no differences observed in the orbitofrontal cortex. Additionally, the mRNA and protein levels of CLDN5, claudin-12, and ZO1 were correlated with the age of onset and duration of schizophrenia [46]. These findings suggest that BBB integrity alterations in schizophrenia might be region-specific rather than widespread across the brain, strongly tying changes in BBB permeability to the pathogenesis of schizophrenia. Studies in Chinese and Iranian cohorts have also identified a slight genetic association between CLDN5 and the development of schizophrenia [99,100]. Specifically, the rs10314 allele has been considered a potential risk factor for psychosis, though this association appears to be population-specific, as it was not confirmed in Japanese populations [101]. Rs10314 is a single nucleotide polymorphism (SNP) that produces a protein of approximately half the mass of the wild-type CLDN5 gene, implying that individuals carrying this allele may produce lower amounts of CLDN5 than typical individuals. The impact of this allele is particularly significant in 22q11 deletion syndrome (22q11DS), where the risk of developing schizophrenia is significantly increased compared to the general population (up to 30% in 22q11DS patients). Studies have shown a significant association between the rs10314 variant and schizophrenia diagnosis in 22q11DS, with these individuals producing only about 25% of the normal CLDN5 levels, potentially leading to increased BBB permeability [102]. The researcher subsequently focused on Pumilio-1 (PUM1), an RNA-binding protein that plays a critical role in cellular processes and is closely associated with the regulation of CLDN5. Studies have shown that under normal physiological conditions, PUM1 functions as a translational enhancer of CLDN-5. However, in rs10314-variant CLDN-5 mRNA, reduced binding affinity with PUM1 leads to translational suppression, resulting in downregulated CLDN-5 expression. In postmortem hippocampal tissues of schizophrenia patients, abnormally elevated CLDN-5 mRNA levels were observed in some individuals, yet the corresponding CLDN-5 protein expression was substantially reduced. A negative correlation was found between CLDN-5 mRNA and PUM1 levels, where patients with high CLDN-5 mRNA levels exhibited relatively lower PUM1 levels. This suggests that in these patients, PUM1’s role in promoting the translation of CLDN-5 mRNA might be hampered, resulting in reduced CLDN-5 protein synthesis [103]. Another study induced patient-derived stem cells to differentiate into BBB cells and found impaired BBB integrity in 22q11DS + schizophrenia patients, with markedly dysfunctional CLDN5 expression observed [104]. Another study evaluated changes in serum zonulin and CLDN5 levels in schizophrenia patients. Compared to controls, schizophrenia patients exhibited significantly higher mean serum zonulin levels and reduced serum CLDN5 levels [105]. In conclusion, the research on tight junction leakage in schizophrenia has uncovered several significant aspects. It is clear that CLDN5 is a crucial protein in BBB tight junctions and its abnormal expression in schizophrenia patients, with variations in different brain regions and associations with disease onset and duration, strongly indicates a link between BBB integrity alterations and the pathogenesis of schizophrenia. The genetic associations identified, particularly the potential role of the rs10314 allele and its interaction with Pumilio-1 in regulating CLDN5 expression, further emphasize the genetic underpinnings of BBB dysfunction in this disorder. However, contradictions still exist. The findings regarding CLDN5 expression are not entirely consistent across different studies, and the population-specific nature of some genetic associations, such as the lack of confirmation in Japanese populations, highlights the complexity of the genetic factors involved. Moreover, while the relationship between CLDN5 and BBB permeability is evident, the exact mechanisms by which changes in CLDN5 lead to BBB disruption and how this impacts the overall pathophysiology of schizophrenia require further in-depth investigation. Future studies should focus on clarifying these uncertainties and exploring potential therapeutic strategies targeting tight junction proteins to better understand and manage BBB dysfunction in schizophrenia.

## 5. The Relationship Between Antipsychotics and Blood–Brain Barrier Function

Common side effects of antipsychotic drugs encompass extrapyramidal side effects, tardive dyskinesia, cardiovascular issues, and metabolic disturbances. Currently, the antipsychotics clinically employed often have suboptimal therapeutic effects and poor bioavailability. However, the effects of antipsychotics on the blood–brain barrier (BBB) are generally twofold. On one hand, given the structure of the BBB in the central nervous system, antipsychotic drugs need to cross the BBB to take effect in the brain. While the BBB protects the central nervous system by limiting the passage and dosage of drugs reaching the brain, it also poses a constraint on the delivery of antipsychotics to the brain. Consequently, the conventional formulations of antipsychotics used in clinical practice are unable to efficiently transport drugs directly into the brain [106]. P-glycoprotein, which is expressed in astrocytes, actively pumps drugs, toxins, and other substances from the BBB back into the bloodstream, preventing harmful substances from piling up in the brain. Certain antipsychotics, like clozapine, can act as P-glycoprotein inhibitors, enhancing drug concentrations across the BBB to a certain extent and exhibiting excellent efficacy in treating schizophrenia. On the other hand, antipsychotics may have an adverse effect on BBB permeability. An in vitro study conducted on the effects of antipsychotics, demonstrated that commonly used antipsychotics, such as clozapine, could induce cytotoxicity and apoptosis in BBB endothelial cells, compromising the barrier function. This might underlie the pathogenesis of antipsychotic-related cerebral edema and neuroleptic malignant syndrome [107].

Moreover, several antipsychotics, such as haloperidol, can reduce levels of inflammatory cytokines, including IL-1β, IL-6, and TNF-α, in schizophrenia patients [108]. Quetiapine, a second-generation antipsychotic, alleviates psychotic symptoms by antagonizing serotonin and dopamine receptors. A study exploring the effects of quetiapine under conditions of high BBB permeability found that quetiapine had a favorable thermodynamically binding activity to MMP-9. In human brain vascular endothelial cells, quetiapine reduced monolayer permeability. The study suggested that quetiapine might possess novel anti-inflammatory properties, and animal models verified its capacity to significantly reduce BBB permeability by maintaining the integrity of tight junctions [109].

Clinical research has demonstrated that incorporating anti-inflammatory agents as adjunctive therapy during the early stages of schizophrenia can be advantageous. Certain antibiotics and nonsteroidal anti-inflammatory drugs (NSAIDs), including minocycline, aspirin, and celecoxib, have shown potential in alleviating negative symptoms and the cognitive deficits associated with schizophrenia [110]. However, the role of the blood–brain barrier in this process requires further exploration.

A variety of strategies have been devised to facilitate drug delivery to the central nervous system (CNS), including intranasal administration [111], the incorporation of nanomaterials [112], RNA interference [113], and extracellular vesicles [114]. However, most of these methods operate under the assumption that a functional blood–brain barrier (BBB) is intact in patients, often overlooking the dynamic characteristics of the BBB and the potential dysfunction of the barrier in psychiatric populations.

## 6. Discussion

The blood–brain barrier (BBB) plays a vital and intricate role in normal brain function, and increasing evidence suggests that BBB disruption is involved in the pathogenesis and progression of diverse neurological disorders. This paper reviews findings based on autopsy studies, serum and cerebrospinal fluid biomarkers, and neuroimaging, indicating that schizophrenia patients display varying degrees of BBB permeability alterations.

Post-mortem studies have spotlighted a series of structural and molecular abnormalities related to BBB function in the brains of schizophrenia patients, such as reduced capillary density and modified molecular marker expression. Peripheral and cerebrospinal fluid markers, including the CSF/serum albumin ratio (QAlb), have provided valuable perspectives on BBB dysfunction. Additionally, neuroimaging techniques, like dynamic contrast-enhanced magnetic resonance imaging (DCE-MRI) and arterial spin labeling (ASL), have enabled in vivo observations of BBB permeability changes, presenting novel viewpoints on the pathophysiology of schizophrenia. These alterations are potentially associated with neuroinflammation, energy metabolism dysfunction, and abnormalities in tight junction proteins.

The principal objective of BBB research in schizophrenia is to clarify its role in pathophysiology and explore its potential as a therapeutic target. Although the existing research has yielded certain outcomes, the research on blood–brain barrier damage in schizophrenia is still in its nascent stage, and prior studies have many limitations. Autopsy-based studies often have small sample sizes and mainly reflect BBB permeability changes in the advanced stages of disease, which might skew the results. Studies based on serum and cerebrospinal fluid biomarkers entail invasive procedures, rendering them less practical for schizophrenia patients, and markers like QAlb and S100b are non-specific, incapable of precisely mirroring BBB permeability changes. Traditional DCE-MRI is highly sensitive to major BBB disruptions and can offer real-time, quantitative, and localized visualization of focal BBB damage [115,116,117]. However, the relatively large molecular weight of exogenous contrast agents (e.g., Gd-DTPA, 550 Da) curtails its capacity to detect subtle early-stage BBB changes. Moreover, concerns regarding the injection of exogenous agents, the risk of nephrogenic systemic fibrosis (NSF), gadolinium deposition and potential neurotoxicity, and prolonged scan times have constrained the application of DCE-MRI in psychiatric research [118]. Another unresolved matter pertains to the temporal relationship between BBB damage and the timeline of schizophrenia development. Few studies have probed the structure and function of the BBB prior to the onset of psychiatric disorders, and there is scant evidence regarding the temporal progression of BBB permeability changes in relation to schizophrenia onset and course. While evidence supports BBB permeability alterations in schizophrenia, further research is requisite to ascertain whether these changes are causal or signify early pathophysiological consequences.

Recent strides in neuroimaging have introduced arterial spin labeling (ASL) perfusion imaging, a non-invasive, quantitative, and spatially resolved technique that utilizes endogenous water as a tracer to assess both cerebral blood flow (CBF) and blood–brain barrier (BBB) permeability [119]. ASL functions by magnetically labeling water protons in arterial blood before they enter the target tissue, facilitating the calculation of key parameters, including the water exchange rate (Kw), the water permeability surface area product (Psw), and CBF. This methodology enables the evaluation of water exchange dynamics across the BBB.

Water, as an abundant endogenous tracer with a relatively low molecular weight, does not freely pass through the BBB. This characteristic allows ASL to detect delicate alterations in BBB integrity during the early stages of disease progression. Consequently, ASL serves as a sensitive and reliable imaging marker for monitoring BBB changes in the initial phases of pathological processes, thus supplying a direct means for tracking BBB dysfunction [120].

Early studies utilizing ASL in schizophrenia have primarily focused on alterations in CBF, both at the global and regional levels. Consistent findings from these studies disclose abnormal CBF patterns in individuals with schizophrenia, such as reduced perfusion in the left superior frontal gyrus, middle frontal gyrus, and right middle occipital gyrus, as well as increased CBF in the left putamen [121]. CBF and the blood–brain barrier (BBB) share a complex interrelationship, jointly contributing to the regulation of neurovascular unit function and neurovascular coupling. Normal CBF is fundamental to maintaining BBB integrity. Abnormal CBF, whether increased or decreased, can lead to changes in BBB permeability. Conversely, elevated BBB permeability can trigger inflammatory responses, which subsequently affect CBF. Inflammation may result in the endothelial dysfunction of cerebral vessels, impairing cerebrovascular autoregulation and disrupting normal CBF patterns. Although changes in CBF are closely associated with abnormalities in BBB permeability and may partially reflect its alterations, this relationship is not absolute. Moreover, CBF changes alone cannot achieve the precise localization or quantitative assessment of BBB permeability alterations [122].

Therefore, to accurately observe changes in BBB permeability, more advanced and targeted approaches are required. To address these limitations, several advanced techniques have been developed. Various ASL-related sequences, such as Intrinsic Diffusivity Encoding of Arterial Labeled Spin (IDEALS), Multi-TE ASL, Diffusion-weighted ASL, Magnetization Transfer Weighted ASL, Contrast-enhanced ASL, and Phase-contrast ASL [123], have been developed and widely applied in studies involving animals, healthy populations, stroke [124], Alzheimer’s disease (AD), small vessel disease [125], obstructive sleep apnea (OSA) [126], multiple sclerosis (MS) [127], old adults [78,79,128], and other conditions [129]. With its inherent sensitivity, short acquisition times, and excellent repeatability, ASL has established itself as a pivotal technique for investigating BBB permeability. Unfortunately, to date, only one study has employed DP-ASL technology to evaluate blood–brain barrier (BBB) permeability in schizophrenia patients, and the study subjects were limited to those who had already received antipsychotic treatment, which somewhat hampers the sensitivity of the ASL technique [77]. A reduction in Kw primarily reflects the impaired water transport functionality of the neurovascular unit, indicating early microscopic alterations in the BBB that are closely associated with the function of aquaporin-4 (AQP4). One study found that aged mice exhibited a 32% increase in BBB water exchange rates, along with a 2.1-fold increase in AQP4 mRNA expression, while the mRNA expression of α-syntrophin, a protein anchoring AQP4 to the BBB, decreased by 7.1-fold [130]. Furthermore, another study comparing AQP4-deficient mice with wild-type mice revealed a 30% reduction in BBB Kw in the AQP4-deficient mice, while no significant differences were observed in arterial transit time (ATT), cerebral blood flow (CBF), or apparent diffusion coefficient (ADC) [131]. These findings underline the sensitivity of ASL in assessing AQP4-mediated BBB water exchange rates.

Neuroinflammation is regarded as one of the significant mechanisms of blood–brain barrier damage in patients with schizophrenia. Astrocytes and AQP4 located on their endfeet play an important role in the occurrence and development of neuroinflammation [132,133]. Previous studies have discovered that AQP4 has the capacity of a neuroimmunological inducer, and it is mainly involved in the occurrence and development of neuroinflammation by promoting astrocyte migration and scar formation, regulating the release of inflammatory factors, and participating in the formation of the regulatory loop of AQP4–astrocyte–inflammatory factor–AQP4 [133,134,135].

Previous research has emphasized the crucial role of AQP4 in various brain pathologies, including schizophrenia [136]. Studies on genetic associations have linked the AQP4 gene to schizophrenia in Southern Han Chinese populations, with risk variants correlating with the severity of negative symptoms and inadequate neuroinflammatory regulation. Furthermore, bioinformatics analyses have identified AQP4 as a critical gene involved in schizophrenia [137]. In addition, recent findings have demonstrated increased AQP4 expression in the prefrontal cortex and hippocampus of MK-801-induced schizophrenia mouse models [138]. Another study based on the MK-801 animal model found that inhibiting AQP4 can reduce the expression of inflammatory cytokines, improve anxiety-like behavior, and alleviate social dysfunction in mice. At the same time, it was found based on bioinformatics analysis thatAQP4 is upregulated in the prefrontal cortex of schizophrenia patients [139]. In schizophrenia model mice, the polarized distribution of AQP4 in the brain is disrupted, and the expression level of AQP4 protein in the model group is significantly reduced compared to the control group. However, in the ursolic acid treatment group, the polarized distribution of AQP4 is improved compared to the model group, and the expression level of AQP4 protein is significantly increased [140]. However, a review of existing research findings reveals that although some progress has been made, these studies are primarily based on animal models and human genetic data, and the results are not entirely consistent. The involvement of AQP4 in the pathophysiology of schizophrenia remains insufficiently understood. Conducting further research into the role of AQP4, especially its contribution to neuroinflammatory processes linked to SCZ, would provide significant insights.

Combining ASL technology with AQP4 research offers a novel approach to understanding and addressing BBB dysfunction in schizophrenia. ASL’s ability to provide real-time, non-invasive assessments of cerebral blood flow and BBB permeability coincides with the need to investigate dynamic changes in AQP4 expression and function. For instance, incorporating ASL-derived metrics such as water exchange rates (Kw) with molecular analyses of AQP4 could yield new insights into the interplay between astrocyte-mediated water transport, neuroinflammation, and BBB integrity.

Clinically, this integrated approach has the potential to improve diagnostic precision and therapeutic strategies. By identifying biomarkers such as altered Kw values and dysregulated AQP4 expression, researchers can better stratify patients and monitor disease progression or treatment efficacy. Additionally, targeted therapies aimed at restoring AQP4 function could mitigate BBB dysfunction and its downstream effects, offering new avenues for intervention.

The present discussion highlights the critical role of blood–brain barrier (BBB) dysfunction in schizophrenia and emphasizes the importance of integrating arterial spin labeling (ASL) and aquaporin–4 (AQP4) research. By combining advanced imaging techniques with molecular insights, researchers are able to uncover the mechanisms underlying BBB disruptions and their contributions to the pathophysiology of schizophrenia. Although challenges remain, particularly in standardizing methodologies and translating findings into clinical practice, this line of inquiry holds great promise for advancing our understanding and treatment of this complex disorder. Specifically, the potential of combining multimodal neuroimaging with biomarkers is discussed to validate underlying mechanisms and explore how these findings can be translated into clinically feasible diagnostic and therapeutic approaches.

## 7. Conclusions

In conclusion, a systematic exploration has been conducted on the intricate role of blood–brain barrier (BBB) dysfunction in schizophrenia, effectively bridging the divide between molecular biomarkers such as aquaporin-4 (AQP4) and advanced neuroimaging techniques like arterial spin labeling (ASL). The intention is to spotlight the theoretical implications of our findings, particularly regarding how BBB disruptions contribute to schizophrenia’s pathophysiology. This work emphasizes the translational prospects of combining AQP4-centered molecular insights with ASL’s non-invasive attributes and calls for ongoing innovation in methodology and technology. Moving forward, future research should prioritize longitudinal studies to clarify the temporal fluctuations of BBB changes and their interplay with neuroinflammation, while simultaneously resolving issues related to standardization and clinical translation. It is believed that this line of research holds revolutionary potential to propel precision medicine, spawn novel diagnostic tools, and devise targeted therapies. By confronting these challenges and building on the theoretical and practical applications of the findings, a solid groundwork can be laid for grasping the intricacy of schizophrenia and enhancing patient outcomes. Overall, this review aims to offer researchers a comprehensive perspective, enhance the understanding of the BBB’s complex role in schizophrenia, and contribute to the development of precision medicine and personalized treatment strategies.

## Figures and Tables

**Figure 1 ijms-26-00873-f001:**
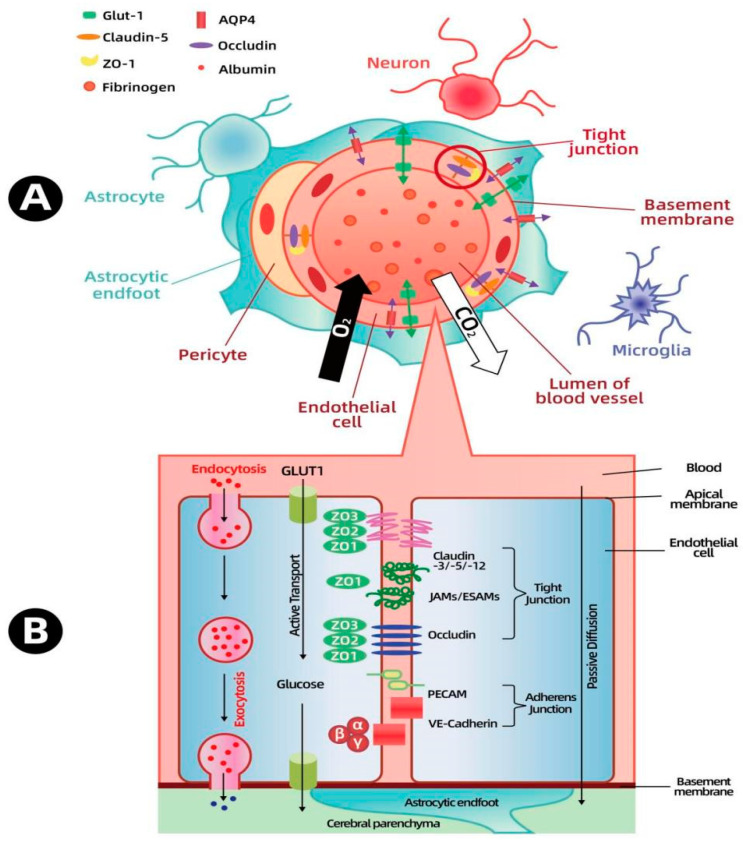
Composition of the neurovascular unit (**A**); substance transport across the blood–brain barrier (**B**).

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
