# Peer review of "Blood–Brain Barrier Disruption in Schizophrenia: Insights, Mechanisms, and Future Directions"

_ijms, 2025, doi:10.3390/ijms26030873_

Round 1
Reviewer 1 Report
Comments and Suggestions for Authors
The article analyzes the state of the blood-brain barrier in schizophrenia from various aspects. The article is written in detail, clearly and orderly. The information in it is presented in an understandable and clear way.
My remarks are two.
1. After each section, it is necessary to make a brief summary of what is clear and what is contradictory. This shows the authors' attitude to the problem. I do not find it in the article.
2. The conclusion is written as an article, not as a conclusion. The presentation of the main conclusions and the summary of all analyzed sections in the article are missing. What is the conclusion and what are the guidelines? What is the take home massage for the reader?
The reviewer
Reviewer 2 Report
Comments and Suggestions for Authors
The manuscript by Zhang et al. provides an insightful overview of the blood-brain barrier's (BBB) role in schizophrenia, covering key topics like structural changes, biomarkers, genetic factors, and neuroimaging techniques. While the content is informative and well-structured, there are several areas of discussion that could be expanded to strength the overall impact:
1. The manuscript does a good job of explaining the roles of astrocytes and endothelial cells in BBB maintenance but falls short to address the complex interplay between NVU components. A more detailed discussion of how these elements interact dynamically and communicate with the peripheral immune system would add valuable depth to the review.
2. Key mechanisms like tight junction disruptions, neuroinflammation, and oxidative stress are discussed individually, but there’s no effort to tie them together into a cohesive framework. The authors should develop an integrated model that explains how these processes collectively contribute to BBB dysfunction in schizophrenia.
3. The authors state the bidirectional effects of antipsychotics on BBB integrity but newer therapeutic strategies such as nanoparticle drug delivery or BBB-targeted peptides are not explored. The authors should include these approaches.
4. The manuscript emphasizes BBB alterations in advanced stages of schizophrenia but doesn’t touch on changes that occur in the early stage. The author should include the early BBB dysfunction and its potential as a predictive biomarker for schizophrenia.
Reviewer 3 Report
Comments and Suggestions for Authors
Journal: IJMS (ISSN 1422-0067)
Manuscript ID: ijms-3402659
Type: Review
Title: Blood-Brain Barrier in Schizophrenia: Current Insights and Future Perspectives
The paper offers a thorough and perceptive analysis of the intricate connection between antipsychotics and the blood-brain barrier (BBB), especially when it comes to schizophrenia. It skillfully illustrates how antipsychotics affect BBB permeability in both directions, outlining the difficulties and possible treatment ramifications. Recent developments in neuroimaging methods, including arterial spin labeling (ASL), are incorporated into the debate to provide a non-invasive way to evaluate the integrity of the blood-brain barrier early in the course of the disease. All things considered, the paper offers a comprehensive viewpoint on the subject, opening the door for additional study and medical developments.
Minor comments:
Introduction:
1. How can studies differentiate between BBB changes as key drivers and secondary phenomena in SZ?
2. Extend the comparison of BBB changes in neurological and psychiatric conditions to elucidate the specificity of SZ findings.
3. Give a more thorough analysis of the processes that underlie BBB alterations and how they interact with environmental and genetic risk factors.
4. Emphasize other possibilities for future study, such as creating tailored treatments or BBB integrity-based diagnostic instruments.
2. Structure and Function of the Blood-Brain Barrier
5. What is the dynamic response of particular BBB components (such as tight junctions and the basement membrane) to stressors like neurotoxins, oxidative stress, or inflammation?
6.What particular impacts does schizophrenia have on tight junctions (such as occludins and claudins) in contrast to other neurological or psychiatric conditions?
7.How does basement membrane degradation affect the early stages of schizophrenia, and can biomarkers predict disease onset or progression?
3. Methodology-Based Studies on Blood-Brain Barrier Permeability in Schizophrenia
8. What differences exist regarding the role of neuroinflammation in BBB malfunction between the various subtypes of schizophrenia, and should the results be interpreted differently for subtypes with high and low levels of inflammation?
5. The Relationship Between Antipsychotics and Blood-Brain Barrier
9. Does the effectiveness of antipsychotics depend on pre-existing BBB modifications, and how do changes in BBB permeability before antipsychotic treatment impact the treatment outcomes?
10. How does BBB function relate to the anti-inflammatory properties of antipsychotics (such as quetiapine and haloperidol) in the context of persistent neuroinflammation in schizophrenia?
11. Given potential risks (such as neurotoxicity and BBB disruption) and the constraints of existing drug delivery methods, what are the practical implications of therapeutically addressing the BBB in schizophrenia?
Reviewer 4 Report
Comments and Suggestions for Authors
The authors present a narrative, expert review on the BBB in schizophrenia. Overall, the review seems to be thorough and exhaustive on the topic, a few points to consider:
1) There is no section on the potential for genetic variation on the BBB and its potential to influence drug activity in schizophrenia. Could this be potentially added?
2) Since this is a narrative review, it is difficult to know what papers or areas are being covered and why. Is it possible to provide some rationale or description on how the review was structured and which papers were selected for review and which for exclusion?
3) in the section on antipsychotics and BBB, you seem to suggest that antipsychotics may not cross the BBB effectively however, wouldn't their massive volume of distribution in addition to the numerous imaging/receptor occupancy studies suggest that they at least have an effect on brain chemistry likely due to their ability to cross the BBB?
